# The Neuropsychology of Emotion and Emotion Regulation: The Role of Laterality and Hierarchy

**DOI:** 10.3390/brainsci11081075

**Published:** 2021-08-17

**Authors:** Oliver Hugh Turnbull, Christian Eduardo Salas

**Affiliations:** 1Centre for Cognitive Neuroscience, School of Psychology, Bangor University, Bangor LL57 2DG, Wales, UK; 2Clinical Neuropsychology Unit, Faculty of Psychology, Diego Portales University, Santiago 8370076, Chile; salasriquelme@gmail.com; 3Centre for Human Neuroscience and Neuropsychology (CEHNN), Faculty of Psychology, Diego Portales University, Santiago 8370076, Chile

**Keywords:** emotion, emotion regulation, process model of emotion regulation, reappraisal, suppression, laterality

## Abstract

Over the last few decades, work in affective neuroscience has increasingly investigated the neural basis of emotion. A central debate in the field, when studying individuals with brain damage, has been whether emotional processes are lateralized or not. This review aims to expand this debate, by considering the need to include a *hierarchical* dimension to the problem. The historical journey of the diverse literature is presented, particularly focusing on the need to develop a research program that explores the neural basis of a wide range of emotional processes (perception, expression, experience, regulation, decision making, etc.), and also its relation to lateralized cortical and deep-subcortical brain structures. Of especial interest is the study of the interaction between emotional components; for example, between emotion generation and emotion regulation. Finally, emerging evidence from lesion studies is presented regarding the neural basis of emotion-regulation strategies, for which the issue of laterality seems most relevant. It is proposed that, because emotion-regulation strategies are complex higher-order cognitive processes, the question appears to be not the lateralization of the entire emotional process, but the lateralization of the specific cognitive tools we use to manage our feelings, in a range of different ways.


*“Neuropsychology is still a very young science, taking its very first steps, and a period of thirty years is not a very long time for the development of any science. That is why some very important chapters, such as motives, complex forms of emotions and the structure of personality are not included in this book. Perhaps they will be added in future editions?”*
Aleksandr Luria (1973), *The Working Brain* [1]

## 1. Introduction

For more than half a century, there has been a debate in neuropsychology on the issue of hemispheric asymmetry in emotion, linked to a broader discussion about the role of cortical brain regions in emotion. The debate has brought data from a wide range of sources: most notably human lesion work [2,3], and electrical stimulation work in non-human animals [4,5]. A key issue in the debate has been whether there is hemispheric asymmetry in the way that the brain processes emotional information in general, a broad question that can be interpreted in several ways. After extensive investigation and discussion, the field now appears to have some resolution to this larger issue. Essentially, there is evidence for hemispheric asymmetry in some elements of emotional life, but not in others. Indeed, the cortex itself is clearly important only for some elements of the broad phenomenon of emotion, for example the way that emotions are perceived and expressed, which often show strong effects of hemispheric asymmetry. As we discuss below, there is also emergent literature describing laterality effects in emotion regulation, closely linked to specific neuropsychological skills. On the other hand, the role of deep subcortical structures in the generation of emotion has become increasingly clear, particularly as regards the experience of powerful feelings, or affect [4,6].

This paper reviews the historical journey of this diverse literature, particularly focusing on the neglected issue of how we manage our emotions, a topic of great importance for mental health and psychopathology. The paper also reviews the status of various emotion-regulation strategies, and the extent to which their neural basis is understood, and proposes three ways in which the field might productively move forward in the future. Our main goal is to expand the limits of the debate regarding emotion and hemispheric asymmetry, pointing to an important but poorly understood emotional process: emotion regulation.

## 2. The Laterality Hypothesis: Perception, Expression, and Experience

The early years of neuropsychological research on emotion were dominated by tasks of emotion perception, and findings on the role of hemispheric asymmetry in people with unilateral brain lesions (see [2] for review). Examples included the role of the right hemisphere in the control of prosody (the intonation and affective aspects that form the ‘music’ of speech), and which so dominates the vocal interactions of mothers and babies [7,8]. There were, for example, related findings of a left ear advantage for the detection of auditory affect, with tasks such as dichotic listening showing strong laterality effects in differentiating (say) between ‘sad’ versus ‘happy’ versions of the same sentence (e.g., [9,10,11,12,13,14]).

An independent strand of research looked at how brain damage could alter the experience of emotion in survivors, and explore the underlying neural and psychological mechanisms. This was particularly studied in patients with denial of deficit (anosognosia) after right-sided brain lesions. These patients often did not show the negative emotional responses to their paresis that one might expect. Most notably, they had fewer episodes of tearfulness and emotional breakdown (so-called ‘catastrophic’ reactions) than those with left hemisphere lesions [15,16,17,18,19,20]. These patients were often not merely unaware of their deficits, but were sometimes unnecessarily optimistic about their medical condition (i.e., ‘euphoric-maniacal’ [18]).

The observation and study of these patients was a cornerstone in the development of one laterality hypothesis suggesting that the ‘absence’ of negative emotions after right-hemisphere damage was related to the impairment of a system specialized in processing emotions of this valence. Evidence from studies looking at the effects of right-sided intra-carotid amobarbital supported the hypothesis. This procedure often produced some degree of anosognosia [21,22,23], long reported to differ from the more emotionally appropriate outcomes that followed from left-sided amobarbital [16] (see [24] for review). A further strand of this line of evidence focused on the role of the right hemisphere in the higher aspects of homeostasis, action, somato-sensory representation, and emotion regulation (pp. 62–69, [25,26]), (pp. 209–213, [27]), [28,29,30,31,32,33].

This pattern of findings led to hemispheric accounts of emotion, of which the best known is that of Davidson and colleagues [34,35]. This is the suggestion of a right frontal system involved in negative (withdrawal-related) emotional states, with left frontal regions associated with positive (approach-related) emotion. Thus, Davidson interpreted depressive reactions in brain-injured patients with left-sided lesions as the result of a disruption of a (left-sided) positive emotion system (e.g., [35] p. 13). In contrast, anosognosia would result from a disruption of a negative emotional system (right-sided), leaving the patient with only a (left-sided) positive emotion system.

## 3. Anosognosia as the ‘Absence’ of Emotion?

There are various limitations to this prototype hemispheric asymmetry model of anosognosia (see [36] for a review). The first is that it fails to account for various forms of emotional complexity in the neurological patient group, including emotion selectivity, and variability across time. For example, a disruption of negative emotions would explain only the absence of emotion in relation to paresis in patients, not why patients might actively deny their paresis [37]. In addition (see [18] for review), the low mood seen in patients with left-sided lesions is likely to result from an emotionally appropriate response to their substantial levels of disability, which typically involves hemiparesis and non-fluent aphasia.

Research on emotion and laterality has also neglected the dynamic character of emotion, particularly emotional experience. During development, human beings learn to manage feelings, particularly painful ones, using either automatic or cognitively controlled regulatory strategies [38,39]. Thus, emotional experience, or emotion generation, cannot be separated from emotion regulation [40], posing important methodological challenges. This fluctuating element is seen in patients with frontal lesions in whom, due to impairments in cognitive control, the dynamics of emotional experience change substantially (e.g., rise time, magnitude, decay rate; see [41,42].

Perhaps an even more complicated question is whether we can even speak of an ‘absence’ of emotion after brain injury. This approach is heavily influenced by how neurology and neuropsychology have historically portrayed cognitive impairments, (correctly) offering accounts of functions that are genuinely lost: a-phasia, a-praxia, a-mnesia, etc. But emotions are not abolished after brain damage. Studies exploring emotional changes after unilateral lesions often report a disruption of specific emotional processes, but not their complete collapse or absence. There are, of course, rare cases in which the processing of a specific emotion can be heavily compromised in all modalities, of which the best known is fear, after bilateral damage to the amygdala [43,44]. However, even in these cases, the loss is not complete, and certainly does not produce an absence of emotional life—not least because other negative emotions are preserved in experience. Case studies that report a preservation of emotional life after extensive bilateral damage to the limbic system support this point, suggesting that neither cortical nor subcortical damage can completely abolish emotional experience [15,34,35,44,45,46,47,48,49,50,51]. Indeed, data from children with hydranencephalic brains, and non-human animals in which the cortex was removed, suggest that the cortex is certainly not the neural substrate of emotional experience, since emotional experience in these cases was preserved and even amplified [5,52,53,54]. Only deep subcortical lesions, especially to the upper brain stem, appear to completely abolish emotional experience, but in this case due to a complete loss of consciousness [55].

There is a further observation that runs counter to the ‘anosognosia as a loss of negative emotions’ account. These are circumstances when right-sided lesions produced an increase, rather than a disruption, in negative emotion (again, [36] for review); for example, the finding of explicit dislike or obsessional hatred of the paretic limb (‘misoplegia’, [56,57]) seen after right-sided lesions, which is discordant with a loss-of-negative-emotions account. These cases often involve damage to right frontal areas of cortex, and several authors have suggested that inhibitory failures contribute to this florid presentation [58]. Finally, there have been reports of frank depression after right fronto-parietal lesions in patients who were also anosognosic (e.g., [58,59,60,61]).

Linked to this, but less cited, are fluctuations in emotion, and awareness of deficit, seen in some anosognosic patients. For example, transient recovery of awareness after caloric irrigation has been reported by various authors [62,63,64,65,66,67,68]). It is of interest that, in at least some of these cases, patients often had a selective failure to recall their (earlier acknowledged) paresis when they had returned to their anosognosic state. Finally, there have been several reports of patients who experienced transient awareness of their deficit, including ‘sudden moments of tearfulness and pre-tearfulness’ [58], p. 166 which appeared to be preceded by themes of loss (cf. [57,60,61,69,70]. This suggests that negative emotions may be intact in at least some instances of anosognosia, appearing only under some circumstances, presumably shaped by cognitive impairments and other dynamic factors that produce variation in emotion in all humans.

As regards the emotional consequences of ideas, we have reported several instances of preservation of negative emotional states, including the full range of emotional experience, in anosognosic patients with right sided-lesions (see [71] for review). An interesting element of these emotional states was their selective nature; for example, a tendency to produce the same magnitude of emotion as controls, but directed at an external object, rather than towards the self, thus suggesting the influence of dynamic or regulatory variables ([31,36,61,72,73], see [71] for review).

A productive explanation of this unusual response is the active influence of some form of psychological defence. In this argument, real-world evidence requires the patient to (consciously) face the existence of their disabling hemiparesis. This triggers powerful feelings of sadness and loss, which are difficult to tolerate. To protect the self from such unmanaged and powerful emotions, the patient produces one or other form of defense (repression, rationalisation, etc., [74]). Defenses are, of course, imperfect solutions to protection, so that the patient will at times be unable to avoid feelings of loss, thus explaining the fluctuating nature of the patient’s emotional state. Arguments of this type have previously been proposed [75,76,77,78,79,80] and have been revived in the neuroscientific literature more recently, suggesting that anosognosia might indeed be a form of defense [37,58,61,65,71,81]. Notably, in the applied field of neuropsychological rehabilitation, it is well established that the clinical presentation of anosognosic patients is a mixture of organic and psychological (defensive) factors [82,83,84,85,86,87].

In sum, we have a field that suggests that emotional experience is somehow preserved after cortical lesions, but is also distorted or altered, in various ways. Explaining this paradox requires firstly that we identify a source of emotion generation that is outside the cortex, and secondly to identify the contribution of neuropsychological components that produce the distortion in emotional processes. Emerging evidence suggests that the source of emotion generation lies in a range of deep subcortical structures, of which the most important is in the upper brain stem. In contrast, the cause of the distortion appears to be a range of cortical areas devoted to managing feelings and other higher cognitive processes. The basis for these assertions is reviewed below.

## 4. The Brain Basis of Emotion Generation

As suggested above, there has been a long-standing debate in neuropsychology on the brain basis of emotion. The alternatives are not merely laterality within the cortex, but the question of whether the basis of emotion generation is itself subcortical. In this anatomical issue, we might include nuclei such as the amygdala, hippocampus, and even archicortical structures such as the insula (e.g., [88], for details on the likely role of these discrete structures in this system in different aspects of emotion, see [5] (pp. 6–8). Critically, this debate also includes deep subcortical structures, such as the hypothalamus, and indeed the upper brain stem, most notably the peri-aqueductal gray (PAG). In part, this debate flourished because the relevant literatures were separated for many decades, notably between human lesion neuropsychology and stimulation work on non-human animals. However, after decades of work, there are several strands of evidence suggesting that the cortex is not the basis of core emotional experience [5,89,90,91,92,93,93,94,95].

Firstly, as touched on above, it has become clear that cortical lesions do not disrupt the ability to generate emotional experiences. Such cortical lesions clearly produce any number of distortions in the emotional lives of neurological patients. These represent a change in the emotional ‘landscape’ (as we might call it): such as an increased threshold to trigger emotional reactions [18], inappropriately positive responses to hemiparesis [16,18], failure to correctly interpret emotions [2], disinhibition of emotional responses [96,97], incorrect use of emotion for decision making [28], or failure to appropriately regulate emotions [98]. However, these cortically lesioned patients preserve the full range of emotional experience: from happiness through anger [71]. The literature has increasingly suggested that the source of emotional experience is deeply subcortical [5,92,93,99].

A parallel research strand has long identified many subcortical emotion-related brain areas, such as the amygdala, insula, hypothalamus, and anterior cingulate (see, for example [81,100,101,102,103]. These seem critical for other elements of emotional life, such as emotion-memory [102,103], the integration of internally generated experienced states with externally facing senses [101], or the role of loss in decision making [100]. However, as described above, lesions to these subcortical sites do not obliterate the emotional experience itself (e.g., [104]). Instead, the core of emotional experience appears to be closely tied to systems underpinning consciousness, in the dorsal regions of the mid-brain [5,92,93], especially the PAG [4,105]. In part, this conclusion is based on evidence that all the primary emotion systems (which include the various subcortical regions named above) terminate in the PAG. In addition, it is in the PAG that one appears to find the maximal emotional outcome (pleasurable or aversive) for the smallest electrical current [4]. Stimulation of the amygdala, striatum, insula, hypothalamus, or anterior cingulate produce fewer substantial effects [4,106,107,108], and lesions to those brain areas produce some, but not overwhelming, changes in global emotional experience [69,70,104,109,110,111,112].

This subcortical source is not the central goal of this review, but it provides a much-needed context for understanding the hierarchical organization of emotional life, in all its potential complexity. Critically, this expands the debate on the neural basis of emotion beyond the problem of hemispheric asymmetry, to the ‘vertical’ dimension of hierarchy. In evolutionary terms, these higher-order cognitive functions have emerged not only to help us successfully deal with demands from the external world, but most importantly to successfully manage internal states of the body (the internal world) in the light of contextual constraints: to manage feelings in an adaptive fashion, in the light of environmental and social limitations. These tools allow us to use emotions to fuel and direct behaviour, to inhibit emotional responses when they are not adequate to our long-term goals, to predict the future based in relevant past experiences, and to read or hide emotional expressions when necessary.

We now further develop this idea, with a focus on the concept of emotion regulation, a complex higher-order psychological process, that has been defined as a mechanism to manage elementary emotional experiences. Below, we describe emerging evidence regarding the neuropsychological and neuroanatomical basis of different emotion regulation strategies, paying special attention to issues of laterality.

## 5. Emotion Regulation

A critical distinction in neuropsychology has been the difference between having feelings (emotion generation) and successfully managing those feelings (emotion regulation). For well over a century, neuropsychologists have noted that brain injury can change the ability to manage feelings (see [98] for review). The Phineas Gage case [96,97] is a commonly cited early example, reporting that the ‘balance’ between Gage’s intellectual faculties and his ‘animal propensities’ had been disrupted. Hughlings Jackson also described the phenomenon as one of alteration of ‘balance’ between cognition and emotion [113] p. 113. The modern literature usually defines these regulation skills as involving a wide set of processes, by which we influence which emotions we have, when we have them, and how they are experienced and expressed [114]. Outside the field of neuropsychology, emotion regulation has been a popular research topic only in the last few decades, and is increasingly linked to a remarkably wide range of mental health disorders [115,116,117,118,119].

Despite the clinical importance of disorders of emotion regulation, the field was relatively under-investigated in neuropsychology for many decades [120]. For example, from 1990 to 2016, only 41 articles were published (roughly 1.5 per annum) that directly addressed the problem of emotion regulation after brain damage [98]. However, the few years since have seen considerable progress in understanding the neurobiological basis of emotion regulation, and in linking this to a robust theoretical framework, namely the well-established Process Model of Emotion Regulation [114]. This model proposes that human beings manage feelings (in a range of ways, from voluntarily to automatic) by using a wide range of regulatory strategies that depend on diverse neuropsychological functions. These are: situation selection, situation modification, attentional deployment, cognitive change, and response modulation. There has been limited investigation of these in neurological populations, and the available evidence is not conclusive regarding laterality effects. Nevertheless, the field is progressively offering more clarity on the issue, and the model offers fertile ground to systematically explore hemispheric asymmetries in the regulation of affect.

### 5.1. Situation Selection

Situation selection is a strongly antecedent-focused emotion-regulation strategy. It has often been described as taking action to avoid/ensure future situations that will produce desirable emotional outcomes, or as the ability to predict the trajectory of emotional experience into the future [121]. An example, sometimes suggested by patients, would be choosing not to go food shopping at a busy time, because this has been stressful in the past. Recent studies have proposed that situation selection might be a particularly useful strategy for those that struggle to manage feelings in the moment [122]. The number of studies exploring this strategy is small, and there are, as yet, no robust experimental paradigms [123,124,125].

Despite its conceptual relevance, and obvious links with mental health problems and emotional difficulties commonly presented by neurological patients, situation selection remains largely under-investigated. It is of particular interest that situation selection appears to depend on two neuropsychological processes: (a) the capacity to generate an array of hypothetical future scenarios, and (b) the ability to decide among them based on their potential emotional impact [98]. Two groups of neurological patients are particularly relevant here. It has been reported that individuals with severe episodic amnesia due to bilateral hippocampal damage [126,127], and individuals with emotion-based decision problems after damage to the ventromedial frontal cortex [27,128], present an impaired performance in tasks that require imagining, and deciding amongst, future scenarios (so-called mental time travel). Regarding laterality effects, none of the above-mentioned studies offered data to address this question systematically.

### 5.2. Situation Modification

Situation modification is a further antecedent-focused emotion-regulation strategy, closer in time to the emotional event. Situation modification occurs in the present moment or immediate future, demanding individuals to rapidly and flexibly generate alternative actions that may change the course of a situation [98]. For example, taking a break from a social situation, such as a group conversation, that is becoming increasingly difficult. Amongst all emotion-regulation strategies, situation modification is the least understood [123]. However, its definition maps into classic conceptualizations of problem solving. We know that a wide range of neuropsychological impairments can compromise problem solving, particularly executive dysfunction. Here, the work of Stuss and colleagues has offered important evidence regarding laterality effects in this regard. Using lesion data, he has proposed a model of executive function in which the left lateral PFC cortex is specialized in task setting and cognitive flexibility, while the right lateral PFC specializes in monitoring behaviour and detecting errors (BA 44, 45, 46, 9, 9/46, and 47/12 [129]). There is some evidence suggesting that individuals with task-setting impairments after left PFC lesions struggle regulating negative feelings, presenting with a tendency to become stuck (to perseverate) in negative emotional states [112]. We are not aware of data regarding emotion-regulation problems in people with lesions to the right lateral PFC. However, monitoring impairments are a cardinal feature of patients with anosognosia [36,130,131].

### 5.3. Attentional Deployment

Attentional deployment involves regulating emotion by changing the focus of attention, moving it away from an aversive object or thought or switching the focus towards a more neutral or positively valenced object or thought. Attentional deployment has been described as an internal version of situation selection [121]. An example might be choosing to focus on a relaxing and uplifting hobby, rather than ruminating on a recent life event that has gone badly. Early during development, and due to the maturation of attentional networks, children learn to redirect attention as a means to regulate negative feelings [132]. Indeed, the experimental study of attentional deployment is a new and small field, and authors have suggested that this strategy relies on several attentional abilities, such as the capacity to sustain attention [133,134], to detach the attentional focus from negative stimuli [132,135] and to inhibit the access of negative material to working memory [136]. Neuroimaging studies looking at attentional deployment/distraction have supported these claims, reporting an increase in activation of fronto-parietal networks related to attentional and inhibitory control [137]. No clear laterality effects have been reported in these studies. Historically, however, the right hemisphere has been described as particularly relevant for many attentional processes. Furthermore, lesions to the right hemisphere generate the most frequent and disruptive attentional deficits [138], while damage to the right PFC has long been known to produce an increase in distractibility [139]. In consequence, there is good evidence, and theoretical arguments [140], to justify the need to systematically explore the relationship between right hemisphere damage, attentional impairments, and the effective use of attentional deployment.

### 5.4. Cognitive Change

Cognitive change broadly refers to the use of verbal thinking to change one’s emotions. It includes by far the best-researched emotion-regulation strategy, namely reappraisal. Reappraisal has been defined as changing a situation’s meaning to alter its emotional impact [115], usually by reframing the meaning of an aversive event in less negative or more positive terms [40,141]. An example might be reframing a relationship break-up by considering the ways in which you might be better off without your (now ex-) partner. Reappraisal has been defined as a complex multi-step process, requiring several neuropsychological components: working memory, inhibition, verbal fluency, and set shifting [98]. In the largest and most complete meta-analysis of reappraisal studies, Buhle and colleagues [142] reported that the implementation of this strategy consistently modulated the amygdala bilaterally, and activated cognitive control regions (dmPFC, dlPFC, vlPFC, and posterior parietal lobe). Of interest to this paper, the authors also described the recruitment of left posterior temporal regions, often implicated in interpreting actions, reflecting on intentions and extracting semantic meaning.

Studies of brain-injured survivors have been a useful source of data to understand the neural basis of reappraisal and its neuropsychological components. In a group study exploring reappraisal generation amongst individuals with unilateral lesions, no laterality effects were found, though there were significant differences between people with and without brain damage [41]. In the same study, we reported that measures of inhibition and verbal fluency were predictors of performance in the reappraisal generation task. As regards anatomy, reappraisal impairments have been regularly described in case studies of individuals with a range of left PFC lesions, consistent with the association between reappraisal and verbal thinking [42,143].

### 5.5. Response Modulation

Response modulation refers to the attempts to influence emotional experience once it has been elicited [114,115]. Two classes of strategy have been proposed to subserve this modulatory skill: *suppression* and *amplification.* Suppression is the best-researched response-modulation strategy. It refers to the inhibition of one’s own emotional expressive behaviour, and is commonly assessed by asking people to watch videos that elicit emotional responses while instructing them to hide their facial expressions. The classic example real-world is that of masked schadenfreude, where it is socially helpful to hide your pleasure in another person’s failure. It has been proposed that suppression requires motor control of facial muscles, response inhibition to withhold a behavioural display triggered by a feeling, and interoceptive/emotional awareness to monitor and adjust the body [98]. Neuroimaging studies have supported this model, reporting both dynamic and structural differences in brain areas classically related to interoception, emotional awareness, and inhibition. Abler and colleagues [144] reported a positive correlation between suppression as an emotion-regulation style and resting-state brain perfusion in the vmPFC (BA 10 and 32). Ohira and colleagues [145] described rCBF increase in the left lPFC (BA 11), mPFC (BA 32), and mOFC (BA 11) during an emotional-suppression task. Goldin and colleagues also reported activation of different PFC regions during a suppression task (right vlPFC, dmPFC, and dlPFC). Studies looking at structural differences have described that individuals that use suppression frequently present larger grey matter volume in the dmPFC [146,147] and insula [148].

Many findings suggest a bilateral involvement of PFC structures, but evidence to support laterality effects is mixed. To our knowledge, there is only one study that has explored the neural basis of suppression in brain-injured survivors with focal lesions. Salas and colleagues [110] used a classic response-modulation paradigm to test whether right PFC damage (commonly linked with inhibition impairments) was associated with emotion dysregulation. Compared to healthy controls, the rPFC group exhibited a reduced range of response modulation: they suppressed and amplified less. Performance on the suppression task was positively associated with suppression usage in everyday life, and motor-inhibition ability. Using lesion-mapping methods, lesion sites of survivors with and without impairment in the suppression task were contrasted, showing that damage to the right posterior insula was the primary common feature of the impaired subgroup.

In sum, there is a small, but rapidly growing, body of literature on the brain basis of emotion regulation. Neuroimaging studies with neurotypical subjects have offered relevant insights, but they are limited in establishing the causal role of this association (see [140]). Research programs such as that of Damasio and colleagues on the role of the vmPFC in decision making are a clear example of the benefits of a multi-method approach [149]. Lesion studies can greatly contribute to this endeavour, and complement neuroimaging data, since they allow us to explore how damage to discrete brain areas are related to specific changes in cognition, emotion, and indeed behaviour. The in-depth study of patients with focal lesions also allows us to capture the subjective experience of those changes, addressing the difficult-to-tackle first- and third-person perspective problem in neuroscience [150]. Importantly, patients with focal lesions can be observed and studied in natural settings, where emotion is at its most powerful, and where emotion regulation is most needed (e.g., [44]). Thus, the field is clearly making progress, but there are many opportunities for improvement, especially as regards integration across methods, and especially in better establishing the role of particular psychological abilities.

As one might expect for such an evolutionarily critical skill, emotion regulation relies on a number of foundational cognitive abilities, and is distributed across wide range of brain areas. In classic neuropsychological terms, emotion regulation is a higher cortical function that depends on the concerted work of widespread cortical, subcortical, and deep subcortical brain areas [151]. This suggests that we should not only consider the historically relevant question of hemispheric laterality, but also the contribution of specific cognitive skills and brain regions. Thus far, there is emerging evidence to support the link between particular emotion-regulation strategies (e.g., reappraisal and suppression) and well-known basic neuropsychological processes (e.g., inhibition and verbal fluency). There are strong theoretical arguments to assume that other less-studied emotion-regulation strategies, such as situation selection and attention deployment, *also* rely on basic neuropsychological processes (e.g., episodic future thinking and attentional control). This is a clear limitation for the field, but also one that can be remedied by additional work, of the sort that has been successful with other strategies (see [98] for review).

Evidence from neuroimaging studies also supports the key role of prefrontal structures in reappraisal and suppression. However, most studies describe bilateral activation of these structures, with few reports describing laterality effects. This is no surprise, if we keep in mind that emotion-regulation strategies are complex, higher-order mental processes, widely distributed and highly dependent on several basic neuropsychological functions. As regards this limitation, we feel that that the study of hemispheric asymmetries in emotion-regulation strategies should focus on the lateralization of the basic component psychological skills, and not on the (more complex) emotion-regulation strategies themselves. For example: language, verbal thinking, and set shifting, which are functions commonly associated with the left lateral convexity, are particularly relevant for reappraisal. Similarly, behavioural inhibition and emotional awareness, frequently associated with the right hemisphere, will be particularly relevant for emotional suppression. In other words, when studying the neural basis of emotion regulation, the relevant question appears to be not the lateralization of the whole process, but the lateralization of the specific cognitive tools we use to manage feelings in different ways.

## 6. Discussion: Three Aspirations

What then are we to make of the current state of our understanding of how we regulate feelings, and the neural basis of this process? Firstly, it is now clear that the core of emotional experience is closely tied to evolutionarily ancient brain systems underpinning consciousness, in the upper brain stem and associated subcortical structures. A range of complex cognitive processes have emerged to help mammals and other vertebrates manage these basic emotional states, in the context of environmental and social constraints (e.g., attaching emotional valence to future events, suppressing or amplifying emotions, or using emotions to frame decisions). Our review has focused on emotion regulation, which represents a diverse set of cognitive control systems to manage these elementary emotional experiences. Given the diversity of psychological processes that we can use to regulate feelings, we find that a diverse set of brain regions are necessary to support this process. These are widely distributed, and suggest hemispheric asymmetry consistent with the lateralization of the basic cognitive processes they rely upon.

With this perspective in mind, we offer three aspirations for the field over the next decade or two. Each is achievable, but of course requires a concerted effort—though fortunately the field has been steadily growing in size and influence.

Firstly, the field needs a more comprehensive model of emotion. There are well-developed elements of emotion science that deal with (say) the experience/generation, perception/ expression, memory, and regulation of emotion. Clearly, these elements operate simultaneously, to try and understand the rounded and complex phenomenon that is emotion. However, research on the specific components operates in largely independent silos. Again, brain-injured patients offer examples to prove that these theoretical silos are artificial constructs. Take, for example, the issue of emotion regulation and emotional reactivity/generation. Both processes tend to be studied separately, despite authors claiming that they are strongly intertwined [152]. It has been widely reported that lesions, particularly to the frontal cortex, can generate an increase in the intensity and magnitude of the emotional response, and a decreased ability to regulate feelings, often referred to as emotional lability, impulsivity, or increased irritability. Here, as noted by Jackson, over a century ago, damage to areas related to cognitive control lower the threshold of emotional reactivity, the well-known ‘short-fuse’ phenomenon that our patients often report. Similarly, patients with damage to brain areas related to the energization system present a decrease in emotional reactivity, in the form of flat affect, apathy, or abulia [129]. In these cases, the threshold to produce an emotional response is too high, so that the down-regulation of emotion is less required, and the amplification of emotion is too taxing. Thus, one key aspiration for the field would be to develop a more solid suite of studies devoted to the interaction between the emotional drivers and the various cognitive components seeking to regulate them.

Secondly, as regards emotion regulation itself, and the issue of laterality, there is a clear need for the field to be more systematic. As discussed above, some emotion-regulation strategies, such as reappraisal and response modulation, have been appropriately investigated after focal brain lesions, using well-designed experimental tasks and self-reporting. This research has identified which neuropsychological processes might underpin each strategy, and it has also shown that there are substantial differences in the brain regions that underpin the strategies. However, the Process Model identifies a wide range of strategies to achieve emotion regulation. As we review above, other relevant emotion-regulation strategies, such as situation selection and attentional deployment, do not have well-designed experimental tasks, and/or have not been investigated in patients with neurological lesions. Here, our efforts should aim at firstly developing the right tools to explore these processes. Insight from neuropsychology itself could prove valuable, since tasks designed to assess specific neuropsychological components could be modified to measure emotion-regulation strategies. The case of episodic future thinking is here a paradigmatic example, with several studies proposing experimental designs to tap this ability [126,153,154,155,156].

Finally, a fully developed model needs to combine both laterality and hierarchy. The field spent many decades framing the emotion question around the cerebral cortex, and around hemispheric asymmetry in particular. As we review in the Introduction, hemispheric asymmetry is an entirely appropriate question, but only in the context of some facets of emotion. Other facets, such as emotional experience or generation, are probably not cortical phenomena at all. In sum, we need a model that encompasses not only the left–right, but also the up–down dimensions of anatomy. This synthesis will be all the easier through work with non-human animals [5]. As discussed above, some facets of emotion (such as core emotional experience, and even emotion memory) are clearly evolutionarily older, and distributed across subcortical areas. Other skills (such as emotion regulation) are evolutionarily newer and achieved by cortical brain regions.

The neuropsychology of emotion is a field that has taken an enormous journey in the last half-century. The study of emotions (as our opening Luria quote suggests) was often absent, or existed as an ‘after-thought’ chapter towards the end of a textbook, based on a modest number of papers, published by a few far-sighted specialists. However, the field was never destined to continue this way, because of the enormous importance of emotions in human mental life, and the critical way that disorders of emotion and their management lie at the heart of mental illness. On this basis, we fully expect that the field will rapidly grow, in both size and influence, and we especially hope that the field moves towards greater precision: to better understand the complex component parts that underpin emotion and emotion regulation.

## Data Availability

Not applicable.

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
