# Peer review of "The Neuropsychology of Emotion and Emotion Regulation: The Role of Laterality and Hierarchy"

_brainsci, 2021, doi:10.3390/brainsci11081075_

Round 1

Reviewer 1 Report

I found the topic of this manuscript interesting and I believe readers of Brain Sciences would also find it of interest. The authors are experts on the topic, the organization is straightforward, and the overall structure of this review is easy to grasp. One minor exception is the very first paragraph before the Introduction; this seems a bit out of place and the statement of neuropsychology having thirty years of development as a science is obviously incorrect since the discipline of neuropsychology has clearly been around before 1990. In fact, there is some sort of reference to Luria (1973) but it's not clear to me why it's here. 

Many topics (or sub-topics) are examined briefly, so in this sense this review if fairly comprehensive. That said, the depth of detail is quite minimal. This is a fairly short review so I assume the goal of the authors was to briefly present or introduce some ideas about brain/emotion relationships, organized around the notion of lateral and/or vertical organizational structures and schemes. Because this journal seems to be very broad in scope with regard to topics, and many readers might not be experts on emotion, one suggestion I have is that it might be helpful to readers if some examples or illustrations are provided. For instance, on page 6 in section 5.1, lines 4- 6. An example illustrating the point of the sentence ending in the Webb et al. (2017) reference, inserting it right after the reference, might make this idea more concrete and understandable to many readers. I would make the same suggestion for the next section (5.2 Situation modification), on line seven of the first paragraph, coming after the sentence ending with "problem solving". And so on. This is less of a critique and more of a simple suggestion. On a formatting note, should Attentional deployment, Cognitive change, and Response modulation be numbered subsections like the first two items of their list of regulatory strategies?

I gave high marks on most categories above, including on whether this article makes a significant contribution to the field, but I admit that while I have some experience with certain issues of depression and apathy as it relates to the neuropsychology of HIV/AIDS (and I am currently on a large international clinical trials team looking at various antidepressant effects on neuropsychological status), I am not a veteran or basic researcher on emotions more generally construed, so I'm not entirely clear on the degree of contribution of this short review. But like I said before, I found it interesting. 

Author Response

We are grateful to the Reviewer for the helpful comments. We are also pleased that the Reviewer found the paper interesting, and appropriate for the Special Issue.

The reviewer’s concerns are addressed in the amended text (in red), and discussed below:

1) "the very first paragraph before the Introduction; this seems a bit out of place"... "some sort of reference to Luria (1973) but it's not clear to me why it's here."

This misunderstanding is a result of a formatting error. In the submitted version of the paper, it opens with a Luria quote. Somehow this was lost when the paper was formatted for the reviewers? The paper itself starts in paragraph 2. Quotation marks have now been added to make this clearer

2) “one suggestion I have is that it might be helpful to readers if some examples or illustrations are provided. For instance, on page 6 in section 5.1, lines 4- 6. An example illustrating the point of the sentence ending in the Webb et al. (2017) reference, inserting it right after the reference, might make this idea more concrete and understandable to many readers. I would make the same suggestion for the next section (5.2 Situation modification), on line seven of the first paragraph, coming after the sentence ending with "problem solving". And so on. This is less of a critique and more of a simple suggestion.

These are helpful suggestions. We have now made amendments to Section 5 to reflect the reviewer’s comments.

3) “On a formatting note, should Attentional deployment, Cognitive change, and Response modulation be numbered subsections like the first two items of their list of regulatory strategies?”

This was our omission. Thank you for noticing it. We have added numbers to the subsections.

Many thanks again for these helpful comments.

Reviewer 2 Report

Reviewer’s comments

The manuscript entitles” The neuropsychology of emotion, and emotion regulation: The role of laterality and hierarchy” aimed to summarize the recent relevant information about neuropsychology of emotions and regulations. The review has lots of significant information and useful for researchers those who are working in emotional behaviors.

The first paragraph before introduction, would be better to be part of the limitation of the present manuscript.

In the introduction, authors would highlight little more information about subtitles of the manuscript. It would increase readers interest.

  1. The laterality hypothesis……., I suggest to included human brain with detailed right and left hemispheres and their emotional connections or circuits. It would help to understand the anatomy. Also, add what is emotional behaviors and examples of negative emotions or positive emotions control by right and left hemispheres respectively.
  2. The brain basis of emotion...…. Why don’t authors divide sub-division of how different brain regions show their emotions like amygdala, PAG. It would help to understand specific brain regions role in the mediation of emotions.
  3. Emotion regulation... Brief introduction need for this chapter.
  4. Discussion is well written. However, please add here in additional paragraph for limitation of this review as well as future perspective.

Major concern.

I noticed that there is no connections between the each sections. Better to connect with upcoming sections, it would make readers to easy to follow up.

Apart from that the review article is interesting with new information.

Author Response

We are grateful to the Reviewer for the helpful comments. We are also pleased that the Reviewer found the paper interesting, and appropriate for the Special Issue.

The reviewer’s concerns are addressed below:

  • “The first paragraph before introduction, would be better to be part of the limitation of the present manuscript.”

This misunderstanding is a result of a formatting error. In the submitted version of the paper, this is a Luria quote, which opens the paper. Somehow this was lost when the paper was formatted for the reviewers? We have added quotation marks to make this clear. The paper itself starts in paragraph 2.

“In the introduction, authors would highlight little more information about subtitles of the manuscript. It would increase readers interest.”

 We have now added some fore-shadowing of the later subtitles in the Introduction.

The laterality hypothesis……., I suggest to included human brain with detailed right and left hemispheres and their emotional connections or circuits. It would help to understand the anatomy. Also, add what is emotional behaviors and examples of negative emotions or positive emotions control by right and left hemispheres respectively.

Unfortunately, the state of the field would not easily allow for a diagram with specified brain areas. We have some anatomical precision for some strategies (reappraisal and response modulation). However, we feel that we could not do justice to a survey of areas in a figure.

“The brain basis of emotion...…. Why don’t authors divide sub-division of how different brain regions show their emotions like amygdala, PAG. It would help to understand specific brain regions role in the mediation of emotions.”

We have added a sentence and reference to clarify the respective roles in Section 4.

“Emotion regulation... Brief introduction need for this chapter.”

We have added a sentence of introduction to Section 5.

“Discussion is well written. However, please add here in additional paragraph for limitation of this review as well as future perspective.”

We have added a few sentences on limitations to Section 5.

“I noticed that there is no connections between the each sections. Better to connect with upcoming sections, it would make readers to easy to follow up.

 We have added connecting sentences where we felt these were needed.

Many thanks again for these helpful comments.